# Soil properties and plant functional traits have different importance in shaping rhizosphere soil bacterial and fungal communities in a meadow steppe

Shanshan Song,[1] Xian Yang,[2] Rong Tang,[1] Yongqiang Zhang,[1] Zhiyao Tang[1]

**ABSTRACT** Soil properties and plant functional traits are important factors influencing rhizosphere microbial diversity and composition. However, their divergent roles in shaping rhizosphere bacterial and fungal communities remain poorly understood. Additionally, the influence of plant resource acquisition strategies on these microbial communities is not well documented. We collected 147 rhizosphere soil, leaf, and root samples from around 18 plant species in a meadow steppe in northern China. We determined bacterial and fungal communities in the rhizosphere soil through high-throughput sequencing. Our analysis revealed that soil properties and plant traits differed in their importance for influencing the diversity and composition of rhizosphere bacteria and fungi. Specifically, soil properties had a more pronounced regulatory effect on bacterial diversity than on fungal diversity. Furthermore, soil properties exerted a stronger influence on the composition of rhizosphere fungal communities compared to plant traits, while both factors similarly affected bacterial community composition. This discrepancy might be attributed to differences in dispersal limitations between bacteria and fungi. We also found that plant resource acquisition strategies significantly impacted both diversity and composition of rhizosphere microbial communities, with plants employing "fast-growing" strategies exhibiting lower fungal diversity. Moreover, conservation-related traits of plants had a more pronounced effect on fungal community composition than root collaboration-related traits. These novel findings demonstrate that soil properties and plant traits play distinct roles in shaping rhizosphere soil bacterial and fungal communities. The linkages between plant resource acquisition strategy and rhizosphere microbial communities could enhance our understanding of the complex interactions between plants and their associated microorganisms.

**IMPORTANCE** Soil microorganisms in the rhizosphere play a crucial role in plant growth and health. However, the specific mechanisms that shape rhizosphere bacterial and fungal communities remain poorly understood. By linking plant resource acquisition strategies to microbial diversity and composition in the rhizosphere, we uncover the pivotal influence of plant conservation traits over root collaboration traits in shaping fungal communities, providing new insights into the complex interactions between plants and their associated microbial communities.

**KEYWORDS** microbial diversity, plant resource acquisition strategy, plant functional traits, rhizosphere soil, soil properties

Soil is the foundation for the establishment of vascular plants and the natural medium for the survival of tens of thousands of microorganisms (1). Defined as the soil surrounding living roots (2), rhizosphere soil differs from bulk soil in microbial diversity and community composition (3). Plants can shape their rhizosphere microorganisms, for

**Peer Reviewer** Christopher W. Schadt, Oak Ridge National Laboratory, Oak Ridge, Tennessee, USA

Address correspondence to Xian Yang, yangx376@mail.sysu.edu.cn.

The authors declare no conflict of interest.

See the funding table on p. 15.

example, by selectively recruiting special protective microorganisms from the bulk soil to suppress pathogens (4); therefore, the microbial diversity in the rhizosphere is typically not greater than that in the bulk soil (3, 5, 6). The significance of such relationships with rhizosphere microorganisms in plant growth and health is well established (7–9). Therefore, understanding the diversity and composition of microbial communities in the rhizosphere is essential for developing strategies to manage and manipulate these rhizosphere microorganisms, ultimately benefiting plant growth and health (10).

It is widely recognized that the composition and diversity of microorganisms in the plant rhizosphere are determined by a combination of factors, including soil properties (11–14), identity and functional traits of host plants (15–17), and root exudates (18–20), among others. The specific soil properties that regulate the composition of rhizosphere microbial community vary across different ecosystems. For instance, the concentration of soil $NH_4^+$–N is a large factor in determining the composition of rhizosphere soil microbial communities in rice and forest soils (21). Soil C:N ratio is critical in shaping the soil microbial communities in subtropical coniferous and broadleaf forest plantations (22), whereas soil C, N, and P concentrations help determine the rhizosphere arbuscular mycorrhizal fungal diversity in forests along an altitude gradient (23). Some glasshouse experiments have also suggested that root morphology and chemical traits are crucial in shaping the composition of rhizosphere microbial communities (24–26).

Undoubtedly, plant functional traits and soil properties collectively influence the diversity and composition of microbial communities. However, the extent to which plant traits and soil properties together explain the composition of the rhizosphere microbiome remains largely unexplored as they are only rarely studied together. A few recent studies have highlighted the important role of soil properties and plant traits in explaining variations in the rhizosphere soil microbial community, both in the alpine *Rhododendron nitidulum* shrub ecosystems (27) and in tropical forests (14). Notably, Hogan et al. (14) reported that soil phosphorus accounts for more variation in the rhizosphere fungal community than plant traits (14). For grassland ecosystems, most existing studies linking plant functional traits to rhizosphere microbial communities have been conducted in monoculture systems (25, 26, 28). The distribution of the fine roots of herbaceous plants, both vertically and horizontally, is generally less extensive than that of trees (29), leading to relatively lower variation in soil properties between adjacent herbaceous plants compared to adjacent trees. It remains an open question whether the impact of soil properties on the diversity and composition of rhizosphere microbial communities in natural grassland communities surpasses that of plant functional traits, as observed in tropical forests ([14], mentioned above).

The "fast–slow" framework of plant resource acquisition presents a spectrum that plants with a "fast" growth strategy typically exhibit high relative leaf and root nitrogen content, high specific leaf area, and low root tissue density, whereas plants on the "slow" end of the spectrum exhibit opposite traits (4, 30). This spectrum extends to their interaction with soil microorganisms: plants with a "fast-growing" strategy tend to have higher root exudation content and faster litter decomposition rates, which stimulate greater microbial activity in the soil compared to plants with a "slow" growth strategy (31, 32). Therefore, plants could regulate the community composition of rhizosphere microorganisms by varying litter quality (33, 34) and root exudation (32). Bergmann et al. (4) proposed the Root Economic Space (RES) framework, which describes two orthogonal axes of root trait variation—the "collaboration gradient" and the "conservation gradient" (i.e., the root "fast–slow" axis) (4). Along the collaboration gradient, plants with thicker and shorter roots are more effective at obtaining soil resources through mycorrhizal fungi (i.e., "outsourcing" acquisition), while plants with thinner and longer roots primarily explore soil resources by themselves (i.e., "do-it-yourself" acquisition) (4, 35). A recent monoculture study on root economics space has shown that saprotrophic and pathogenic fungal community composition was affected by root conservation and collaboration gradient, respectively (28). However, plants in natural grasslands constantly adjust their traits due to competition for resources, and how plant resource acquisition

and conservation strategies, as well as collaborative strategies, are associated with the diversity and composition of rhizosphere microbial communities remains unclear.

Additionally, most studies on rhizosphere microorganisms have focused on fungal communities (14, 26, 28), with comparatively fewer considering both bacteria and fungi (36, 37). Soil bacteria and fungi are substantially different regarding their growth habitats, dispersal abilities, and interactions with plants. On one hand, the symbiotic relationships between certain fungi, such as mycorrhizal fungi, and plant roots are primarily influenced by specific plant traits (38, 39). Thus, plant traits might play a more crucial role in shaping fungal community structures, especially in symbiotic relationships, than they do in bacterial communities. On the other hand, filamentous fungi are larger and more dispersal limited, in contrast with unicellular bacteria that are smaller and have stronger dispersal abilities (37, 40). Plants employ distinct resource acquisition strategies to modulate root exudates, thereby creating unique rhizosphere habitats that serve as ecological filters (41). The strong dispersal ability of bacteria in soil (42) enables the plant rhizosphere to recruit bacteria from a wider area of bulk soil. Therefore, in this context, plant traits might have a more pronounced effect on the structure of bacterial communities than on fungal communities. Soil properties have often been shown to play the largest role in shaping the composition of bacterial and fungal communities (11, 43, 44). However, as fungi generally have lower dispersal capacity than bacteria (37), soil properties may have a particularly strong effect on the composition of slowly dispersing fungal communities.

In this study, we explored the influence of soil properties and plant functional traits on the diversity and composition of rhizosphere microbial communities in a meadow steppe of northern China. We collected 147 rhizosphere soil samples from 18 plant species across three 1-ha sites. The rhizosphere soil bacterial and fungal communities were determined through high-throughput sequencing. Additionally, we measured soil properties at each site and assessed leaf and root functional traits of plants corresponding to the collected soil samples, aiming to determine their relative impact on the structure of rhizosphere bacterial and fungal communities. Meadow steppes are characterized by a variety of herbaceous plants and beneficial conditions of soil moisture and fertility. In such environments, the diverse and dense root systems of herbaceous plants provide a substantial source of nutrients for bacteria, consequently influencing their diversity and community structure. In contrast, fungi, particularly those dependent on mycelial networks, are more likely to be more influenced by the soil's physical and chemical properties, such as organic matter content, moisture, and pH levels. Therefore, we hypothesized that the composition and diversity of rhizosphere bacterial communities are more influenced by plant functional traits than by soil properties, whereas the composition and diversity of rhizosphere fungal communities are more prominently determined by soil properties than by plant functional traits. We also hypothesized that the conservation gradient exerts a greater impact on the composition and diversity of rhizosphere bacterial communities, whereas the collaboration gradient exerts a greater impact on the composition and diversity of rhizosphere fungal communities.

## MATERIALS AND METHODS

### Study site and design

The study was conducted in a meadow steppe at Saihanba Station (42°18′N, 117°16′E, 1,544 m a.s.l.), located in Hebei Province, China. This region has a temperate semi-humid continental monsoon climate, with a mean annual temperature of 0.2°C and a mean annual precipitation of 438 mm, based on data from 2021 to 2022, the year prior to, and the year of, sampling. The soil type in this area was identified as chestnut soil (45). The growing season of alpine grassland lasts from May to September. The main species at the research site include *Leymus chinensis* (grass), *Carex korshinskyi*, *Carex lanceolate* (sedges), *Medicago ruthenica* (legumes), *Sanguisorba officinalis*, *Artemisia tanacetifolia*, and *Plantago asiatica* (non-legume forbs).

Sixteen hectares of grassland was fenced off in 2020 to facilitate grassland monitoring. In 2022, we selected three 1-ha sites (site A, B, and C). In each site, we randomly set three plots, each with an area of 20 m × 20 m and a distance of at least 20 m between them.

## Plant leaf and root functional traits

Based on the average relative cover of species, we classified the species into three categories: dominant species (with a relative cover >5%), common species (1%–5%), and rare species (<1%). We selected all five dominant species and randomly chose nine and four species from the common and rare species, respectively, as target species (Fig. S1). In each plot, we randomly selected at least three whole plants without obvious pathogen infection from the 18 target species and excavated these entire plants to measure their functional traits. To ensure the integrity of the root systems during sampling, we first conducted destructive sampling outside the designated plots to approximate the depth range for each species' root systems. In the sampling plots, we excavated the soil mass using a shovel, maintaining a distance of at least 20 cm from the target plant to minimize disturbance. The excavation depth was guided by the preliminary data obtained outside the plots. After that, we carefully extracted the entire root system of each target plant from the surrounding soil mass, ensuring that the roots were fully separated and intact. Due to the sparse distribution of some plant species in certain plots, we were unable to collect all planned samples and ultimately obtained 147 plant individuals.

We then pooled the above- and below-ground parts of these plants and measured a total of 21 plant functional traits, covering key morphological and chemical characteristics of leaves and roots, as well as two community-level traits (46). Leaf traits included leaf thickness (LT), specific leaf area (SLA), leaf dry matter content (LDMC), leaf mass per area (LMA), leaf carbon content (LC), leaf nitrogen content (LN), leaf phosphorus content (LP), and leaf carbon-to-nitrogen ratio (LCN). Root traits comprised specific root area (SRA), specific root length (SRL), root tissue density (RTD), average root diameter (RD), branching intensity (BI), specific root tip abundance (SRTA), root carbon content (RC), root nitrogen content (RN), root phosphorus content (RP), root carbon nitrogen ratio (RCN), and root depth (D). In addition, we recorded average plant height (height) and relative cover (cover) at the plot level, as community-level functional traits (Table S1).

To measure leaf saturated weight, we soaked the leaves in distilled water for at least 24 h, then gently blotted them dry with absorbent paper before weighing. We used a Vernier caliper to measure the thickness of at least three leaves and averaged them as LT and used a scanner (Epson V39) to get the area of the leaves. The scanned leaves were dried at 65°C for 48 h to obtain their dry weight. We calculated the SLA as the leaf area per unit dry weight, LDMC as the ratio of leaf dry weight to leaf saturated fresh weight, and LMA as the leaf dry weight per unit area. We used an element analyzer to measure the LC and LN, and used the molybdenum antimony anti-colorimetric method to determine LP (47).

We randomly selected at least five root samples for each species within each plot to determine root functional traits. We removed soil attached to the root system using deionized water before measuring the root traits. Then, we dissect the root order based on the morphometric approach (48, 49). We used the fully branched lateral roots to measure root morphological traits (50) and considered the most distal root tip as first order. Because the first two or three orders of dicotyledons are absorptive roots, while all orders of monocotyledons are absorptive roots (50), only the first two- or three-order roots were collected for the measure of the morphological traits. We used a scanner (Epson V850 Pro) to get the root system images and then used the image-processing software WinRHIZO PRO 2004b (Regent Instruments Inc., Quebec, QC, Canada) to obtain the total length, total surface area, total volume, and root tip number. We dried the scanned root in the same way as the leaves to obtain the root dry weight. We calculated the SRA as the ratio of the total area to root dry weight, SRL as the ratio of the total length to root dry weight, RTD as the ratio of the root dry weight to total volume, BI as

the ratio of the root tip number to total length, and SRTA as the number of tips per unit of dry weight. We used the same methods as those of the leaves to measure the RC, RN, and RP.

## Soil measurements

We collected three soil samples along the diagonal position of each plot, and each sample was obtained by mixing three drills of soil within a 1-m$^2$ area. We used fresh soil to measure the soil water content (SWC), ammonium nitrogen ($NH_4^+$–N), and nitrate nitrogen ($NO_3^-$–N). Fresh soil samples were dried in an oven at 105°C for 24 h to determine the SWC. We measured $NH_4^+$–N and $NO_3^-$–N by 2 mol•L$^{-1}$ KCl extraction–indophenol blue colorimetry and dual-wavelength method and measured pH, soil organic carbon (SOC), total nitrogen (TN), total phosphorus (TP), and available phosphorus (AP) for the air-dried soil. We used an elemental analyzer (FLASHEA 112 Series, Thermo Electron, USA) to measure the SOC and TN, and the water extraction method with a 1:5 soil–water volume ratio to measure the soil pH. We used the molybdenum–antimony anti-spectrophotometric and anti-colorimetric methods to determine TP and AP, respectively (47). Since we did not record the distances between the plants and the soil sample collection point within each plot, we averaged the soil properties across the three samples for each plot.

## DNA extraction and sequencing of rhizosphere soil

After excavating the roots, we removed the loose non-rhizosphere (bulk) soil from the root system and scraped off the rhizosphere soil adhering to the root surface with sterile tweezers. A minimum of 0.5 g of fresh rhizosphere soil samples was collected for each species within each plot. We extracted soil DNA using the MagaBio Soil/Feces Genomic DNA Purification kit (Hangzhou Bioer Technology Co., Ltd.) and measured the concentration and purity of the DNA samples using NanoDrop One (Thermo Fisher Scientific, MA, USA). For bacterial community analysis, the V4–V5 regions of the bacterial 16s rRNA gene were amplified using the specific primers 515F (GTGCCAGCMGCCGCGGTAA) and 909R (CCCCGYCAATTCMTTTRAGT) with a 12-bp barcode unique to each sample (51). For fungal community analysis, the ITS2 regions were amplified using the specific primers gITS7 (GTGARTCATCGARTCTTTG) and ITS4 (TCCTCCGCTTATTGATATGC), also with a 12-bp barcode unique to each sample (52). We used 1% agarose gel electrophoresis and GeneTools Analysis Software (Version 4.03.05.0, SynGene) to detect the length and concentration of the PCR products, respectively, and then pooled them in equimolar concentrations. Finally, we purified the mixture of PCR products using E.Z.N.A. Gel Extraction Kit (Omega, USA). The 2 × 250-bp paired-end sequencing strategy was used to sequence the library by Illumina Nova6000 platform (Guangdong Magigene Biotechnology Co., Ltd. Guangzhou, China).

## Bioinformatics analysis

We removed the primers using the CUTADAPT software (https://github.com/marcelm/cutadapt/) to obtain paired-end clean reads. These reads were spliced using USEARCH -fastq_mergepairs (http://www.drive5.com/usearch/) to generate raw tags. The quality control of the raw tags was performed using the fastp package (version 0.14.1, https://github.com/OpenGene/fastp). Specifically, the raw tags were processed using a sliding window approach with a window size of four bases and a mean quality score threshold of 20 (Q20). Sequences were then clustered at 97% similarity using UPARSE, producing an OTU (operational taxonomic unit) clustering table. Taxonomic annotation of the OTUs was conducted with the USEARCH-sintax command, and OTUs with fewer than 10 total counts were removed. We assembled rarefaction curves (Fig. S2) to assess the sequencing depth and determine appropriate thresholds for sample inclusion. Based on the rarefaction analysis, samples with low sequencing depth or high levels of noise were excluded. We then retained 141 bacterial and 146 fungal samples and rarefied their sequences to 41,922 and 21,425 reads, respectively, to standardize sampling effort.

## Statistical analysis

All statistical analyses were conducted using R (version 4.3.2) (53). We calculated bacterial and fungal OTU richness and Shannon diversity using the vegan package (54). We employed one-way analysis of variance (ANOVA) to test differences in bacterial and fungal diversity across sites or species. The same approach was used to assess differences in the relative abundance of different phyla taxa, as well as plant functional traits and soil properties across sites or species. *Post-hoc* Tukey HSD comparisons were performed using the "TukeyHSD" function from the stats package (53). We constructed linear mixed-effects models to explore the effect of plant functional traits on rhizosphere soil microbial diversity, with OTU richness and Shannon diversity as response variables, plant functional traits as fixed explanatory variables, and plots nested within sites as random intercepts. Additionally, we constructed multiple regression models to assess the relative importance of individual variables on rhizosphere soil microbial diversity. Prior to analysis, each variable was standardized (average = 0, SD = 1) to facilitate comparison of the regression coefficients. The "forward.sel" function from the adespatial package was used to select the optimal variables affecting rhizosphere soil microbial diversity (55). We also constructed multiple regression models to assess the relative importance of sites and species on rhizosphere soil microbial diversity. Pearson correlation analysis was employed to test the relationships between the relative abundance of rhizosphere microbial communities at the phylum level and environmental factors (i.e., soil properties or plant functional traits).

To explore the effects of plant resource acquisition strategies on rhizosphere soil microbial diversity, we performed principal component analysis (PCA) separately for plant leaf and root traits. For each set of traits, we selected the principal components that cumulatively explained more than 50% of the total variance. These components were used as plant scores along the conservation and collaboration strategy gradients. We analyzed the relationship between these strategy scores and rhizosphere soil microbial diversity using linear mixed-effects models, with plots nested within sites as random intercepts.

To visualize the differences in microbial community composition at the OTU level across sites or species, we used non-rarefied, Hellinger-transformed OTU read numbers (28, 56) and conducted principal coordinates analysis (PCoA) based on Bray–Curtis dissimilarity. We used permutational multivariate analysis of variance (PERMANOVA) to separately test the significance of differences in microbial community composition between sites and species and employed the "pairwise.adonis" function from the pairwiseAdonis package to evaluate pairwise differences in microbial community composition (57). To explore the relative contributions of soil properties and plant functional traits to bacterial and fungal community composition, we first conducted distance-based redundancy analysis (db-RDA) using the "capscale" function in the vegan package (54). Prior to the analysis, we selected non-redundant variables via forward selection ("forward.sel" function in the adespatial package, 999 permutations) (55) and log-transformed these selected variables. To quantify the individual contributions of each variable, we performed hierarchical partitioning using the "rdacca.hp" function from the rdacca.hp package (58). To further explore how plant resource acquisition strategies affect the composition of bacterial and fungal communities, we performed another db-RDA using conservation and collaboration scores as explanatory variables. In this analysis, plot was included as a conditional variable to account for plot-level variation. We then fitted generalized additive models (GAM) using the "ordisurf" function in the vegan package to visualize the relationships between strategy scores and microbial community composition (54).

We performed variance partitioning analysis (VPA) based on multiple regression on dissimilarity matrices (MRM) to explore the relative contributions of soil properties, plant functional traits, and species phylogenetic distance to the compositional differences of bacterial and fungal communities. The phylogenetic distance between plant species was calculated using the "cophenetic" function from the ape package (59), based on the

phylogenetic tree constructed with the "phylo.maker" function from the V.PhyloMaker2 package (60). We also conducted a Mantel test with 999 permutations to assess the effect of spatial distance on bacterial and fungal Bray–Curtis dissimilarity between samples, using the "mantel" function from the vegan package (54). Note that this analysis only considered distances between plots, and spatial distances within plots were approximated as zero.

## RESULTS

We identified a total of 11,397 OTUs for rhizosphere soil bacteria and 4,120 OTUs for fungi across the three sites. The top 10 abundant bacterial phyla were Proteobacteria (28.8%), Acidobacteria (16.6%), Actinobacteria (14.5%), Bacteroidetes (14.3%), Planctomycetes (7.2%), Chloroflexi (5.6%), Verrucomicrobia (4.8%), Thaumarchaeota (2.7%), Gemmatimonadetes (1.5%), and Armatimonadetes (0.5%). The top five abundant fungal phyla were Ascomycota (59.6%), Basidiomycota (23.8%), Mortierellomycota (6.3%), Glomeromycota (1.5%), and unidentified (7.7%) (Fig. S3 and S4).

### Variations of soil properties and plant functional traits

Soil water content ($F_{(2,24)} = 6.19$; $P = 0.007$), available phosphorus ($F_{(2,24)} = 6.483$; $P = 0.006$), ammonium nitrogen ($F_{(2,24)} = 5.936$; $P = 0.008$) and nitrate nitrogen ($F_{(2,24)} = 6.935$; $P = 0.004$) varied significantly among sites (Fig. S5). No significant differences in soil pH ($F_{(2,24)} = 3.237$; $P = 0.060$), soil organic carbon ($F_{(2,24)} = 1.051$; $P = 0.365$), total nitrogen ($F_{(2,24)} = 0.253$; $P = 0.778$), and total phosphorus ($F_{(2,24)} = 0.792$; $P = 0.464$) were found among sites (Fig. S5). All plant functional traits, except for leaf carbon content, differed significantly between species (Fig. S6).

### Composition and diversity of rhizosphere microbial community

The OTU richness and Shannon diversity of bacteria, as well as the OTU richness of fungi, varied significantly among sites, with site C exhibiting significantly lower diversity than sites A and B (Fig. S7a through c). Additionally, the Shannon diversity of both bacteria and fungi, along with the OTU richness of fungi, differed significantly among plant species (Fig. S8).

The composition of bacterial and fungal communities exhibited significant variations, both between sites and among different plant species (Fig. 1; Fig. S9; Table S2). Focusing on the most dominant phyla, the relative abundance of Proteobacteria and Thaumarchaeota, within bacterial communities, varied significantly among sites and plant species

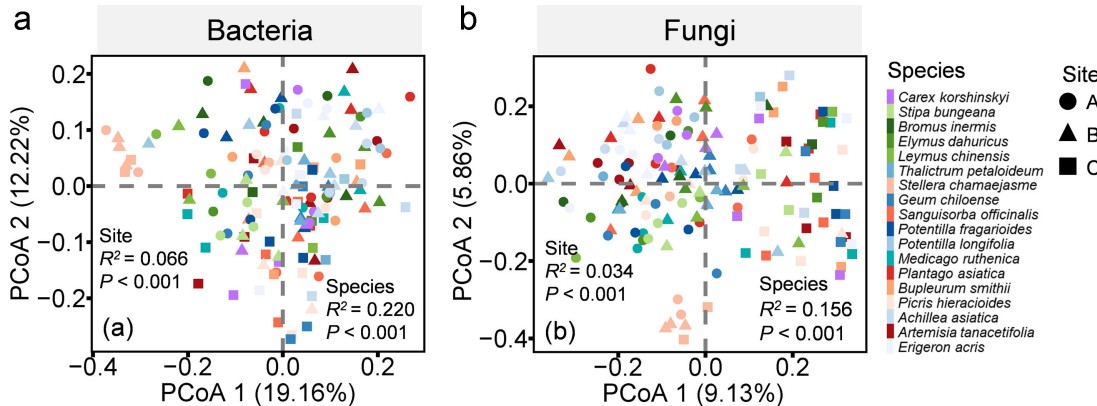

FIG 1 Principal coordinates analysis (PCoA) of the differences in bacterial (a) and fungal (b) community composition between samples, based on the Bray–Curtis dissimilarity. Different-colored and -shaped points represent rhizosphere soil samples collected in different plant species and sites, respectively. $R^2$ and $P$: PERMANOVA test statistic.

(Fig. S4; Table S3). Among the dominant fungal phyla, the relative abundance of Glomeromycota varied significantly among sites and plant species (Fig. S4; Table S3).

## Effects of soil properties and plant functional traits on rhizosphere microbial diversity

The mixed-effects models and multiple regression models showed that root depth negatively affected the OTU richness and Shannon diversity of bacteria, while specific leaf area and root nitrogen content negatively affected OTU richness and Shannon diversity of fungi (Fig. 2; Fig. S10). Soil properties (77%) played a more important role in driving the OTU richness of bacteria compared to plant functional traits (23%) (Fig. 2). Plant functional traits (63%, 67%, 100%) played a more important role in driving the Shannon diversity of bacteria, the OTU richness, and Shannon diversity of fungi compared to soil properties (37%, 33%, 0%) (Fig. 2). Site (58%) played a more important role in driving the OTU richness of bacteria compared to species (42%) (Fig. S11). Species played a more important role in driving the Shannon diversity of bacteria, the OTU richness, and Shannon diversity of fungi compared to site (Fig. S11).

PCA of leaf traits showed that PC1 represents a conservation gradient for leaves (hereafter $PC1_{Leaf}$), with higher scores indicating a "fast-growing" strategy (Fig. 3a; Table S4). For root traits, SRL, SRTA, and SRA exhibited strong positive loadings on the PC1 axis, while RD showed a strong negative loading. This axis thus represents a root collaboration gradient ($PC1_{Root}$) from "outsourcing" to "do-it-yourself" strategies (Fig. 3b; Table S4). On the PC2 axis, RCN and RTD showed positive loadings, whereas RP and RN had negative loadings, representing a root conservation gradient ($PC2_{Root}$). Higher scores on $PC2_{Root}$ are associated with a "slow-growing" strategy (Fig. 3b; Table S4). We note that RTD, a conservative trait in Bergmann et al.'s RES framework (4), also showed a moderate loading on PC1, suggesting that this axis may partially reflect variation in tissue density. The Shannon diversity of bacteria was negatively correlated with $PC2_{Root}$ (Fig. 3h). The OTU richness and Shannon diversity of fungi were negatively correlated with $PC1_{Leaf}$ (Fig. 3c and f), and the OTU richness and Shannon diversity of fungi were positively correlated with $PC2_{Root}$ (Fig. 3g).

## Effects of soil properties and plant functional traits on rhizosphere microbial composition

The variables selected by db-RDA with forward selection explained approximately 17% of the variation in bacterial community composition ($F_{(9, 132)} = 3.10$, $P < 0.001$, $I_{total} = 11.73$, $I_{constrained} = 2.05$, $I_{unconstrained} = 9.69$) and 25% for fungal communities ($F_{(14, 131)} = 3.11$, $P < 0.001$, $I_{total} = 33.34$, $I_{constrained} = 8.31$, $I_{unconstrained} = 25.03$) (Fig. 4; Table S5). The large SWC difference between site C and the other two sites was the most influential factor for both bacterial and fungal communities. The results of hierarchical partitioning indicated that SWC contributed 38% of the variation for bacteria and 14% for fungi. Root depth was the most significant plant functional trait affecting the composition of these communities, accounting for 18% of the variation in bacteria and 8% in fungi. Overall, plant functional traits (49%) and soil properties (51%) affect community composition of bacteria to a similar extent. Soil properties had more contributions to the variation in community composition of fungi (73%) than plant functional traits (27%). The results of the Mantel test showed that spatial distance exerted a stronger influence on Bray–Curtis dissimilarity of fungi ($r = 0.224$, $P < 0.001$) compared to that of bacteria ($r = 0.433$, $P < 0.001$; Fig. 4e and f). The results of MRM-based VPA indicated that species phylogenetic distance provides a weaker explanation for bacterial and fungal communities than soil properties and plant functional traits (Fig. S13). $PC1_{Leaf}$, $PC1_{Root}$, and $PC2_{Root}$ were strongly associated with the community composition of bacteria and fungi (Fig. 5).

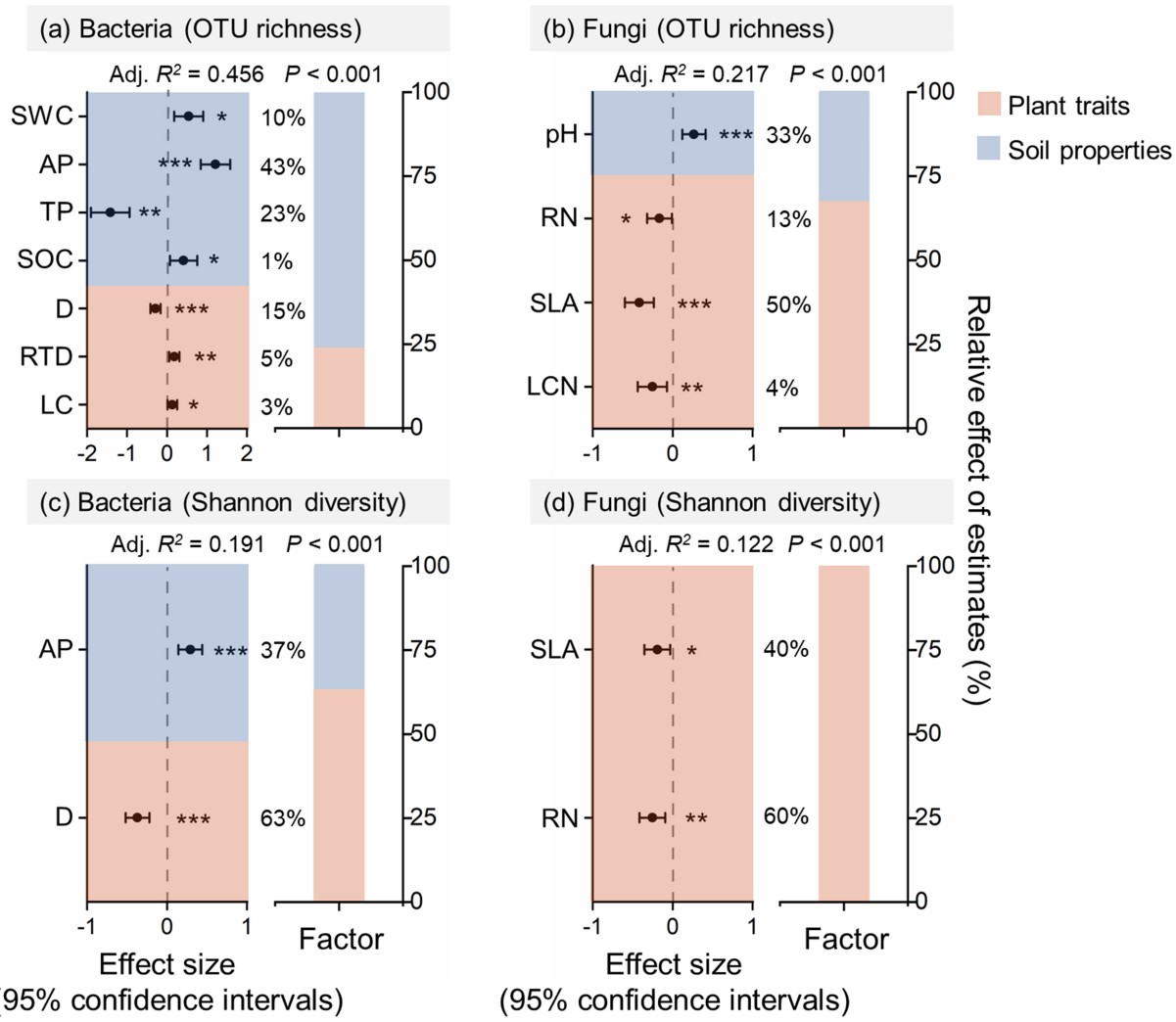

**FIG 2** Results of multiple linear regressions of the effects of soil properties and plant functional traits on soil bacterial (a, c) and fungal (b, d) diversity, and their relative contributions. Effect sizes are standardized coefficients for each predictor variable. Solid and open circles indicate significant and insignificant effects, respectively. *$P < 0.05$; **$P < 0.01$; ***$P < 0.001$. RN, root nitrogen content, D, root depth; RTD, root tissue density; LC, leaf carbon content; LCN, leaf carbon nitrogen ratio; SLA, specific leaf area. A detailed description of each soil property and plant functional trait is presented in Table S1.

## DISCUSSION

Soil properties and plant functional traits are crucial factors in determining diversity and composition of rhizosphere microbes, yet their specific roles in natural grassland ecosystems remain inadequately explored. Our study reveals two novel insights. First, we found that soil properties and plant traits vary in their importance for shaping the composition of rhizosphere bacteria and fungi. Notably, soil properties have a greater impact on the composition of rhizosphere fungal communities compared to plant traits, likely due to the more limited dispersal of fungi, while both factors similarly influence bacterial community. Second, we show that plant resource acquisition strategies play a significant role in structuring rhizosphere microbial communities, with "fast-growing" strategy plants showing reduced fungal diversity. Additionally, plant traits along the conservation gradient exert a stronger influence on fungal community composition than those along the root collaboration gradient.

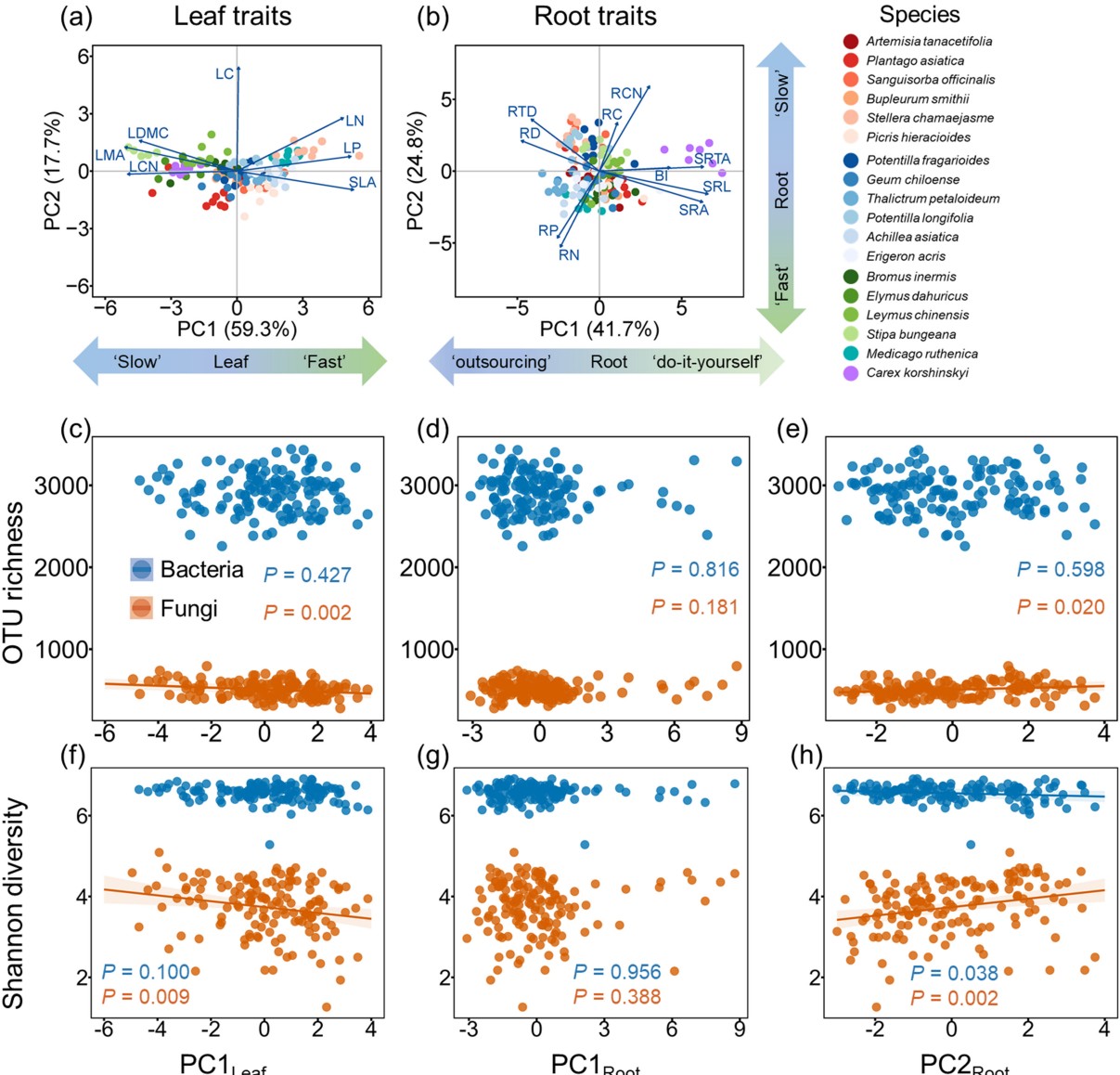

**FIG 3** (a, b) Principal component analysis (PCA) of plant functional traits. Trait abbreviations: RCN, root carbon nitrogen ratio; RC, root carbon content; RP, root phosphorous content; SRTA, specific root tip abundance; BI, branching intensity; RD, average root diameter; RTD, root tissue density; SRL, specific root length; SRA, specific root area; LN, leaf nitrogen content; LP, leaf phosphorous content; LMA, leaf mass per area; LDMC, leaf dry matter content. Detailed descriptions of all traits are provided in Table S1. (c through h) Scatterplots illustrating the effects of species scores along plant functional trait gradients on rhizosphere microbial diversity, based on linear mixed-effects models. $PC1_{Leaf}$ represents the leaf conservation gradient (from "slow" to "fast" strategy), $PC2_{root}$ represents the root conservation gradient (from "fast" to "slow" strategy), and $PC3_{Root}$ represents the root collaboration gradient (from "outsourcing" to "do-it-yourself"). Panels (c)–(e) show OTU richness, and panels (f)–(h) show Shannon diversity. Solid lines indicate significant ($P < 0.05$) relationships, and the shaded areas represent the 95% CI of the fit.

## Influences of soil properties and plant functional traits on the diversity of rhizosphere soil microbial communities

Soil properties had a stronger effect on rhizosphere bacterial diversity than on fungal diversity, which differs from our original hypothesis regarding diversity (Fig. 2). This finding aligns with the results reported by Hao et al. (61), indicating that bacterial diversity is more responsive to variations in soil properties (61). Most soil properties could predict rhizosphere bacterial richness, with soil available phosphorus (AP) being the most significant factor. The strong positive correlation between bacterial diversity and soil AP is consistent with findings from studies on phosphorus fertilization and its

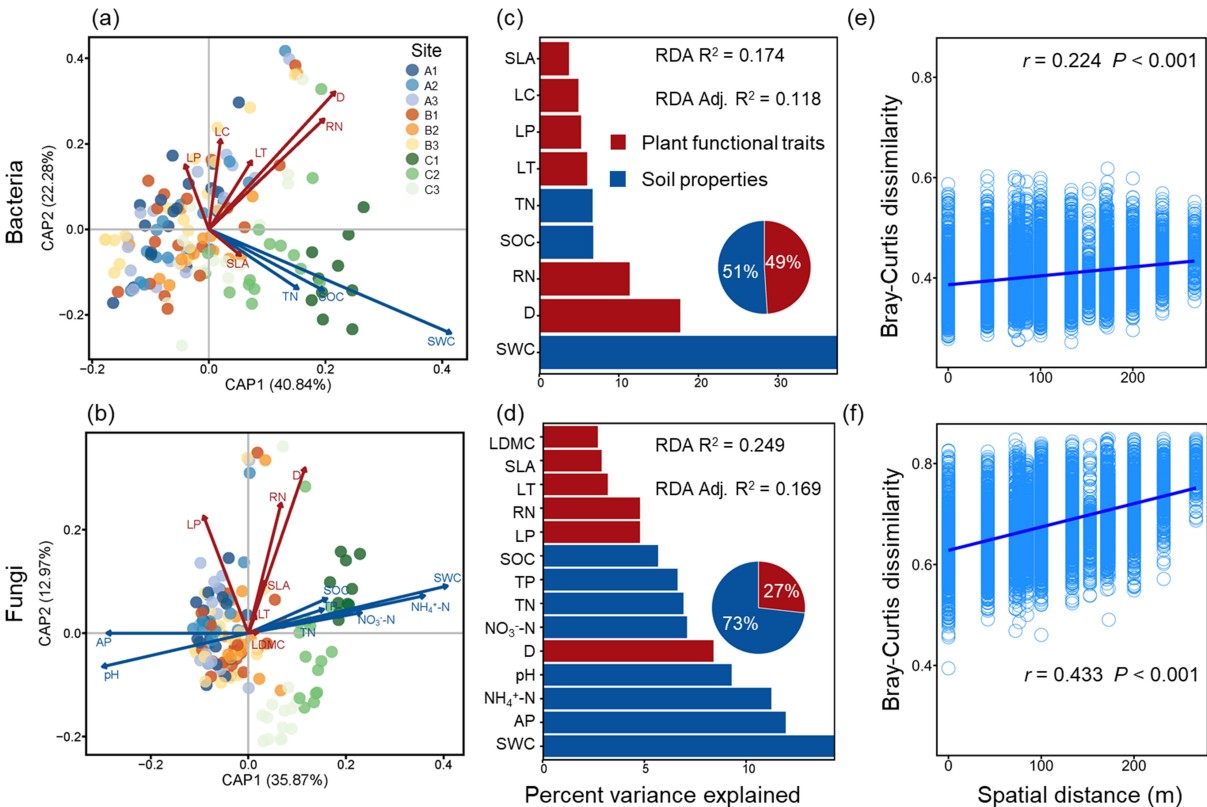

**FIG 4** db-RDA of rhizosphere bacterial (a) and fungal (b) communities. Different colored points represent rhizosphere soil samples collected in different sites. Soil properties and plant functional traits are represented as blue and red arrows, respectively. The numbers after CPA1 and CPA2 represent the rate of explanation of the variable by axis 1 and axis 2, respectively. The relative contribution of soil properties (blue) and plant functional traits (red) to the dissimilarity in bacterial (c) and fungal (d) communities was determined by hierarchical variation partitioning of the variables in db-RDA. Relationships between Bray–Curtis dissimilarity of rhizosphere microbial community composition and spatial distance (e, f). Mantel tests were used to test their relationships (with 999 permutations), and $r$ values shown in the plot are Mantel coefficients. Solid lines indicate significant correlations ($P < 0.05$).

effects on soil bacterial diversity (62, 63). Additionally, it is also possible that increased bacterial diversity in the rhizosphere enhances the dissolution of inorganic phosphorus, as demonstrated by Zhou et al. (64) in their study on soybean rhizosphere bacteria (64). In contrast, fungal richness was predicted only by soil pH, indicating that fungi are less sensitive to a broader range of soil properties. The positive effect of soil pH on rhizosphere fungal richness is consistent with the results of Labouyrie et al.'s (2023) study on soil microbial diversity across Europe (65). We also found that soil pH was negatively correlated with the abundant phylum Mortierellomycota, while no other abundant phyla showed this pattern, indicating that only a few fungal taxa could maintain high abundance in acidic soils (Fig. S12).

Both morphological and chemical traits of plants could affect the diversity of rhizosphere microbes (66). We found that root depth, a key morphological trait indicating the distribution of plant roots within the soil profile, negatively affected rhizosphere bacterial diversity (Fig. 2). This pattern might be attributed to the comparatively lower diversity of bacterial seed banks in deeper soil layers, as microbial diversity in bulk soil typically peaks at the surface and decreases with increasing depth (67). Factors contributing to this decrease in diversity include reduced soil enzyme activity, limited availability of plant-derived carbon, and changes in soil properties, all of which may lead to the loss of bacterial diversity in deeper soil layers (68, 69). It is also possible that reduced microbial diversity at depth leads to decreased enzyme activity, which in turn affects nutrient cycling. In plants with deep root systems, like *Stellera chamaejasme* (Fig. S6), the rhizosphere soil primarily comes from root tips distributed in deeper layers, potentially

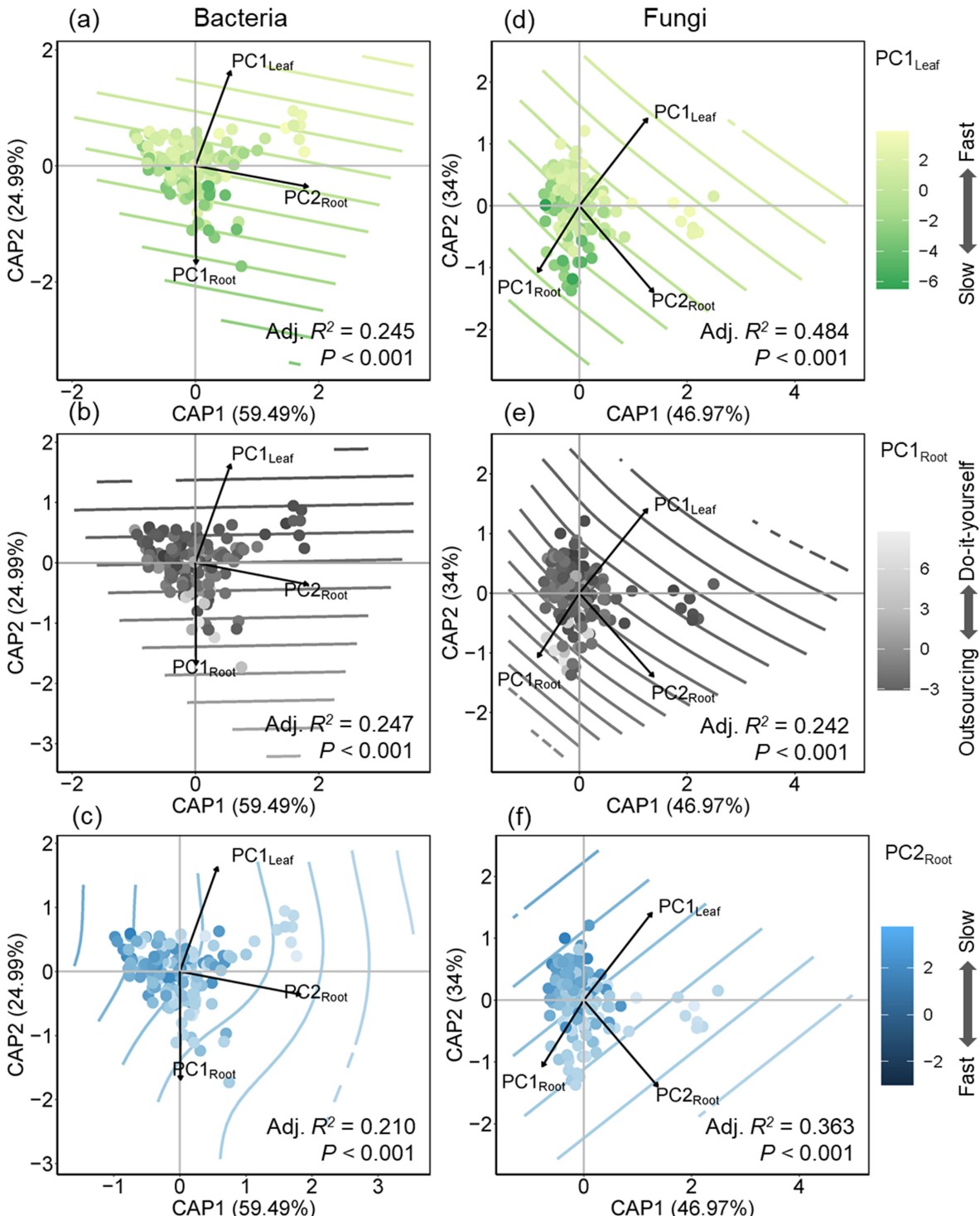

**FIG 5** db-RDA of rhizosphere bacterial (a, b, c) and fungal (d, e, f) communities. Different colored points represent the scores in conservation (PC1$_{Leaf}$: the conservation gradient of leaf from "slow" to "fast", PC2$_{Root}$: the conservation gradient of root from "fast" to "slow") and collaboration (PC1$_{Root}$: the collaboration strategy of root from "outsourcing" to "do-it-yourself") gradients. The numbers after CPA1 and CPA2 represent the rate of explanation of the variable by axis 1 and axis 2, respectively. Colored lines show fitted values for PC1$_{Leaf}$, PC1$_{Root}$, or PC2$_{Root}$ from generalized additive models.

resulting in lower bacterial diversity in their rhizosphere. Note that the lack of stratified sampling and the relatively small sample size in this study may limit the detection of depth-related patterns. In addition, our results suggested that root depth does not significantly affect fungal diversity within the rhizosphere, in contrast to its impact on

bacterial communities, indicating that soil bacteria are more sensitive to soil depth than fungi. This finding is consistent with that of Ko et al. (70), who reported that soil depth has a more pronounced effect on the composition of bacterial communities compared to that of fungal communities in bulk soil (70).

Plant chemical and morphological traits may influence rhizosphere fungal diversity either directly by affecting the extent of symbiotic fungal colonization in roots or indirectly by regulating the composition of root exudates (71, 72). Plants at the "fast" end of the leaf and root economic spectrum, typically characterized by high SLA, LN, and RN, and low LDMC and RTD (32, 73), produce a greater rhizosphere priming effect through root exudates, such as organic acid (74), which is related to their high leaf photosynthetic and root resource acquisition capacities (75). Such root exudates can potentially reduce microbial diversity through mechanisms such as lowering soil pH (76, 77). Contrary to our initial hypothesis, we found that the conservative gradient, rather than the collaboration gradient, had a greater impact on rhizosphere fungal diversity. Specifically, plants employing acquisitive ("fast-growing") strategies exhibit lower fungal diversity compared to those with more conservative ("slow-growing") strategies (Fig. 3). This finding aligns with that of Hennecke et al. (78), who reported that lower saprophytic fungal diversity in "fast" root trait plant communities may result from bacteria's superior ability to utilize labile carbon compounds (78), but contrasts with that of Hennecke et al. (28), who found that root trait axes did not strongly influence rhizosphere fungal diversity in monoculture grassland (28). This discrepancy may be due to the greater variability in plant functional traits caused by interspecific competition in natural grasslands, resulting in more pronounced differences in the rhizosphere microhabitats of different species.

We note that the root collaboration axis ($PC1_{Root}$) also captured some variation in conservative traits, such as RTD, which could have influenced the observed relationship between the conservation gradient and fungal diversity. Additionally, we also note that RD and RTD, commonly associated with collaboration and conservation strategies, respectively, were positively correlated in our data set. This overlap may indicate that the two gradients are not entirely independent in trait space, which is not uncommon in natural systems, and should be considered when interpreting their relative contributions to microbial communities.

## Contribution of soil properties and plant functional traits to the variation of bacterial and fungal communities

Soil properties are important environmental factors affecting the composition of rhizosphere microbial communities, as highlighted in previous studies (21, 23). Among these soil properties, we found that SWC notably influences both bacterial and fungal communities, corroborating the study of Monokrousos et al. (79), which reported the direct impact of the water regime on rhizosphere microbes in areas with low SWC (79), comparable to the levels observed in our study region. Lower soil moisture not only limits the spread of nutrients, but also affects the transport of bacteria and fungi in the soil, as well as their uptake of soil nutrients (80).

Among various functional traits, root depth was the most significant factor explaining variations in the rhizosphere microbial communities, especially in bacterial communities (Fig. 4). Considering the complexity of the plant root systems, it is challenging to ascertain the root depth of different species and soil properties of their rhizosphere. Eilers et al. (67) found that soil depth significantly impacts bacterial communities (67), particularly in the relative abundance of Bacteroidetes and Verrucomicrobia, which decrease and increase with soil depth, respectively. Aligned with this, our results showed that the relative abundance of these bacterial phyla was negatively and positively correlated with root depth, respectively (Fig. S12). Although our study reveals a potential link between root depth and the composition of rhizosphere soil microbial communities, this finding is limited by methodological constraints. In particular, the absence of stratified sampling across soil depths and limited replication hinder our ability to distinguish the effects of root depth from those of soil depth and other covariates.

RN emerged as another important root trait affecting the composition of rhizosphere microbial communities, particularly bacterial communities (Fig. 4). This influence may be due to the close relationship between RN and the root exudate metabolome, as suggested by Williams et al. (41).

Differing from our hypothesis, we found that the conservation ("fast–slow") gradient had a greater impact than the collaboration gradient on the composition of rhizosphere fungal communities (Fig. 5). While the previous study by Zeng et al. (81) identified root morphological traits representing the "collaboration gradient" (81), such as root branching intensity, as key factors shaping rhizosphere soil microbial communities, our study found that root morphological traits failed to explain the variation in the composition of rhizosphere microbial community (Fig. 4), and the relationship between root collaboration gradient and relative abundance of most phyla was not significant (Fig. S12). This suggests that root traits associated with the conservation gradient, which may be linked to root exudation, could have a more substantial regulatory effect on rhizosphere fungal communities compared to traits related to the collaboration gradient, which are often associated with mycorrhizal fungi in resource acquisition. One possible explanation for this is that mycorrhizal colonization is typically restricted to specific microbial taxa, whereas root exudates can broadly influence the growth and distribution of a wider range of microbial groups (82, 83).

Consistent with our hypothesis, we found that soil properties contribute more to the composition of rhizosphere fungal communities (73%) than plant functional traits (27%) (Fig. 4). However, for bacteria, soil properties (51%) and plant traits (49%) are of nearly equal importance (Fig. 4). Note that unmeasured plant traits and soil properties may also influence the results. While the plot-scale soil parameters employed in this study may not capture microscale heterogeneity within the rhizosphere, they effectively reflect edaphic variation at the plant community level. Notably, all measured soil properties significantly influenced rhizosphere fungal composition, suggesting that fungal communities are primarily influenced by the surrounding soil environment. The greater increase in fungal community dissimilarity with spatial distance compared to bacteria (Fig. 4) supports the idea that this difference between bacteria and fungi might be related to weaker dispersal ability in fungi. Dispersal is recognized as a significant factor in shaping both bacterial and fungal communities in soil (84, 85), with fungi being more constrained by dispersal limitation than bacteria (37, 86). In the rhizosphere, bacteria's greater dispersal ability allows them to be recruited to the plant rhizosphere from a wider range of bulk soil. Consequently, plants with different resource acquisition strategies may influence bacterial composition in the rhizosphere by secreting specific exudates that create conditions favorable for certain beneficial bacteria, leading to their increased abundance around the roots (87, 88). This process results in a greater influence of plant traits on bacterial communities than on fungal communities, where fungi, due to their relatively weaker dispersal ability, are more influenced by soil properties.

## Conclusion

This study is the first, to our knowledge, to distinctly highlight the divergent roles of soil properties and plant functional traits in shaping rhizosphere soil bacterial and fungal communities within natural grassland ecosystems. By linking plant resource acquisition strategies to microbial diversity and composition in the rhizosphere, we uncover the pivotal influence of plant conservation traits over root collaboration traits in shaping fungal communities, providing new insights into the complex interactions between plants and their associated microbial communities. However, our conclusions are based on amplicon sequencing with a single set of primers at one time point, which may introduce biases in the microbial community detection. Additionally, the lack of stratified rhizosphere sampling and the use of soil properties measured at the plot level may have limited our ability to capture fine-scale variation in microbial responses. Future studies employing multiple primer sets and a broader temporal sampling design could help reduce such biases and improve the detection of a wider spectrum of

microorganisms. Incorporating depth-specific rhizosphere sampling and higher-resolution soil measurements may also help assess the generality of our findings and further advance our understanding of the ecological processes structuring plant-associated microbial communities.

## ACKNOWLEDGMENTS

This work was supported by the National Key Research and Development Program of China (Grant number 2022YFF0801801), the National Natural Science Foundation of China (Grant numbers 32025025 and 32101273), and the Basic and Applied Basic Research Foundation of Guangdong Province (Grant number 2022A1515012068).

Conceptualization: S.S., X.Y., and Z.T.; methodology: S.S., X.Y., R.T., Y.Z., and Z.T.; formal analysis and investigation: S.S., R.T., and Y.Z.; writing—original draft preparation: S.S. and X.Y.; writing—review and editing: S.S., X.Y., and Z.T. All authors have read and approved the final manuscript.

## AUTHOR AFFILIATIONS

[1]State Key Laboratory for Vegetation Structure, Function and Construction (VegLab), Institute of Ecology, and College of Urban and Environmental Sciences, Peking University, Beijing, China
[2]State Key Laboratory of Biocontrol, School of Ecology, Sun Yat-sen University, Shenzhen, China

## AUTHOR ORCIDs

Shanshan Song  http://orcid.org/0000-0003-3189-5196
Xian Yang  http://orcid.org/0000-0002-1527-7673

## FUNDING

| Funder | Grant(s) | Author(s) |
| --- | --- | --- |
| National Key Research and Development Program of China | 2022YFF0801801 | Zhiyao Tang |
| National Natural Science Foundation of China | 32025025 | Zhiyao Tang |
| National Natural Science Foundation of China | 32101273 | Xian Yang |
| Basic and Applied Basic Research Foundation of Guangdong Province | 2022A1515012068 | Xian Yang |

## AUTHOR CONTRIBUTIONS

Shanshan Song, Conceptualization, Formal analysis, Investigation, Methodology, Writing – original draft, Writing – review and editing | Xian Yang, Conceptualization, Methodology, Writing – original draft, Writing – review and editing | Rong Tang, Formal analysis, Investigation, Methodology | Yongqiang Zhang, Formal analysis, Investigation, Methodology | Zhiyao Tang, Conceptualization, Methodology, Writing – review and editing

## DATA AVAILABILITY

Raw Illumina sequence data are deposited in the Sequence Read Archive (SRA) of the National Center for Biotechnology Information (Bacteria: PRJNA1067179; Fungi: PRJNA1067330). All other data that support the findings of this study are available on Dryad Digital Repository (https://doi.org/10.5061/dryad.bg79cnpn8).

## ADDITIONAL FILES

The following material is available online.

## Supplemental Material

**Supplemental material (mSystems00570-25-s0001.docx).** Table S1–S5; Fig. S1–S13.

## Open Peer Review

**PEER REVIEW HISTORY (review-history.pdf).** An accounting of the reviewer comments and feedback.

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
