## [Reviewer comments · mSystems]

Soil properties and plant functional traits have different importance in shaping rhizosphere soil bacterial and fungal communities in a meadow steppe

Shanshan Song, Xian Yang, Rong Tang, Yongqiang Zhang, and Zhiyao Tang

Corresponding Author(s): Xian Yang, Sun Yat-sen University - Shenzhen Campus

Review Timeline:

Submission Date:	April 21, 2025
Editorial Decision:	May 13, 2025
Revision Received:	May 23, 2025
Accepted:	June 4, 2025

Editor: Juliana Almario

Reviewer(s): Disclosure of reviewer identity is with reference to reviewer comments included in decision letter(s). The following individuals involved in review of your submission have agreed to reveal their identity: Christopher W. Schadt (Reviewer #1)

Transaction Report:

DOI: <https://doi.org/10.1128/msystems.00570-25>

Re: mSystems00570-25 (Soil properties and plant functional traits have different importance in shaping rhizosphere soil bacterial and fungal communities in a meadow steppe)

Dear Dr. Xian Yang:

To support future research, please make sure that all the data is available:

-Please verify that the data deposited in NCBI is available (for example the Bacteria data : PRJNA1067179, is not available)

-The data in the Dryad Digital Repository does not seem to be accessible, please adjust (http://datadryad.org/share/aNqSB_YvTXzbgavR4CX8liZZcgVo_71IMBCP0oiDCJ4)

-Please upload the trait data to the FRED and TRY databases and please indicate the link.

Revision Guidelines

Sincerely,
Juliana Almario
Editor
mSystems

Reviewer #1 (Comments for the Author):

I reviewed a previous version of this manuscript. I greatly appreciate that the authors were very responsive to the suggestions of both myself and the other reviewer: revising text, discussion, methods and performing new analyses where requested and documenting it well in the R2R text. They have also streamlined the manuscript discussion and focus by omitting prior, less essential, parts that made it somewhat confusing to follow at times and moving some material to the supplement.

I have no further requested changes.

Reviewer #2 (Comments for the Author):

I want to thank the authors for carefully revising the manuscript. I am glad to see that changes in statistical analyses did not strongly alter the conclusions, demonstrating some robustness of the results. My only remaining concern is that the close correlation of RD and RTD that might make 'outsourcing' and 'slow' strategies less independent than the authors currently suggest. I do agree with the author's interpretation in L. 402 that PC2 largely reflects a fast-slow conservation gradient. As such, the overall approach of using PC2 scores to test how this tradeoff affects microbial communities is valid. However, the close correlation of RD and RTD (the two most common proxies for 'outsourcing' and 'slow' strategies in the literature) indicates that these two strategies are maybe not completely independent and effects of PC1 (which are currently regarded as effects of collaboration only) might be to a lesser degree also due to changes in tissue density. I would suggest to: 1. Mention the discrepancy between RTD and other proxies of 'slow' strategies (RCN, RC) and hence also the deviation from the Bergmann-RES and 2. Add a caveat that due to this, RTD might partly contribute to the effects of the collaboration gradient on microbial communities.

Apart from this I only have very minor text edits listed below:

- L. 286: please cite R (as done previously) and Rstudio only additionally
- Fig. 4c, d: Either decide for "Percent variance explained" OR "R2"
- Fig. 2: Effect size, not Effective size

Thanks for revising this nice manuscript!

Point-by-point responses to reviewers' comments

Note: texts in black are the comments, and texts in blue are our responses.

Editor:

To support future research, please make sure that all the data is available:

-Please verify that the data deposited in NCBI is available (for example the Bacteria data: PRJNA1067179, is not available)

Response: We have verified that the Bacteria data with the accession number PRJNA1067179 deposited in NCBI is now available.

-The data in the Dryad Digital Repository does not seem to be accessible, please adjust (http://datadryad.org/share/aNqSB_YvTXzbgAVr4CX8liZZcgVo71IMBCP0oiDCJ4)

Response: We have provided publicly available links in the Data availability section (Lines 623-624):

“All other data that support the findings of this study are available on Dryad Digital Repository (<https://doi.org/10.5061/dryad.bg79cnpn8>).”

-Please upload the trait data to the FRED and TRY databases and please indicate the link.

Response: We noticed that the FRED and TRY databases share overlapping datasets, and TRY has already integrated most of the functional trait data from FRED. To avoid redundancy, we have opted to deposit our trait data exclusively in the TRY database and make it publicly available upon completion of the review process. However, due to the uncertain timeline of TRY's curation process, the original description of uploading data to FRED/TRY has been removed from the manuscript. We have confirmed that the raw data for this study has been publicly available in the Dryad Digital Repository (<https://doi.org/10.5061/dryad.bg79cnpn8>).

Reviewer #1 (Comments for the Author):

I reviewed a previous version of this manuscript. I greatly appreciate that the authors were very responsive to the suggestions of both myself and the other reviewer: revising text, discussion, methods and performing new analyses where requested and documenting it well in the R2R text. They have also streamlined the manuscript discussion and focus by omitting prior, less essential, parts that made it somewhat confusing to follow at times and moving some material to the supplement.

I have no further requested changes.

Response: Thank you for your positive feedback.

Reviewer # 2 (Comments for the Author):

I want to thank the authors for carefully revising the manuscript. I am glad to see that changes in statistical analyses did not strongly alter the conclusions, demonstrating some robustness of the results. My only remaining concern is that the close correlation of RD and RTD that might make 'outsourcing' and 'slow' strategies less independent than the authors currently suggest. I do agree with the author's interpretation in L. 402 that PC2 largely reflects a fast-slow conservation gradient. As such, the overall approach of using PC2 scores to test how this tradeoff affects microbial communities is valid. However, the close correlation of RD and RTD (the two most common proxies for 'outsourcing' and 'slow' strategies in the literature) indicates that these two strategies are maybe not completely independent and effects of PC1 (which are currently regarded as effects of collaboration only) might be to a lesser degree also due to changes in tissue density.

I would suggest to: 1. Mention the discrepancy between RTD and other proxies of 'slow' strategies (RCN, RC) and hence also the deviation from the Bergmann-RES and 2. Add a caveat that due to this, RTD might partly contribute to the effects of the collaboration gradient on microbial communities.

Response: Thank you for your careful review and constructive comments. We fully acknowledge your concern that the close correlation between RD and RTD may challenge the independence of the 'outsourcing' (PC1) and 'slow' (PC2) root strategies. To address this, we have made the following revisions:

(1) As suggested, we have clarified in the Results section that although PC1 primarily reflects the root collaboration gradient, with strong positive loadings of SRL (0.944), SRTA (0.905), and SRA (0.898), and a strong negative loading of RD (-0.680). RTD, a conservative trait in the RES framework proposed by Bergmann et al., also showed a moderate negative loading on PC1(-0.594), suggesting that this axis may partially capture variation in tissue density as well (Lines 401-409).

(2) We have added a caveat in the Discussion noting that RTD, a conservative trait, showed moderate loading on both PCA axes. Its overlap with RD suggests some trait-level correlation between the collaboration and conservation gradients, which may influence the interpretation of their respective effects on microbial communities (Line 514-521).

Results:

“For root traits, SRL, SRTA, and SRA exhibited strong positive loadings on the PC1 axis, while RD showed a strong negative loading. This axis thus represents a root collaboration gradient ($PC1_{Root}$) from “outsourcing” to “do-it-yourself” strategies (Figure 3b; Table S4). On the PC2 axis, RCN and RTD showed positive loadings, whereas RP and RN had negative loadings, representing a root conservation gradient ($PC2_{Root}$). Higher scores on $PC2_{Root}$ are associated with

a “slow-growing” strategy (Figure 3b; Table S4). We note that RTD, a conservative trait in Bergmann et al.’s RES framework (4), also showed a moderate loading on PC1, suggesting that this axis may partially reflect variation in tissue density.” Lines 401-409

Discussion:

“We note that the root collaboration axis ($PC1_{Root}$) also captured some variation in conservative traits such as RTD, which could have influenced the observed relationship between the conservation gradient and fungal diversity. Additionally, we also note RD and RTD, commonly associated with collaboration and conservation strategies, respectively, were positively correlated in our dataset. This overlap may indicate that the two gradients are not entirely independent in trait space, which is not uncommon in natural systems, and should be considered when interpreting their relative contributions to microbial communities.” Lines 514-521

Apart from this I only have very minor text edits listed below:

- L. 286: please cite R (as done previously) and Rstudio only additionally

Response: Thank you for your correction. Please refer to Lines 288:

“All statistical analyses were conducted using R (version 4.3.2) (53).”

- Fig. 4c, d: Either decide for "Percent variance explained" OR "R2"

Response: Thank you for your correction. We have modified it to “Percent variance explained”.

Please refer to Figure 4c, d.

Figure 4. Distanced-based redundancy analysis (db-RDA) of rhizosphere bacterial (a) and fungal (b) communities. Different colored points represent rhizosphere soil samples collected in different sites. Soil properties and plant functional traits are represented as blue and red arrows, respectively. The numbers after CPA1 and CPA2 represent the rate of explanation of the variable by axis 1 and axis 2, respectively. The relative contribution of soil properties (blue) and plant functional traits (red) to the dissimilarity in bacterial (c) and fungal (d) communities were determined by hierarchical variation partitioning of the variables in db-RDA. Relationships between Bray-Curtis dissimilarity of rhizosphere microbial community composition and spatial distance (e, f). Mantel test were used to test their relationships (with 999 permutations), and r values shown in the plot are Mantel coefficient. Solid lines indicate significant correlations ($P < 0.05$).

- Fig. 2: Effect size, not Effective size

Response: Corrected. Please refer to Figure 2.

Figure 2. Results of multiple linear regressions of the effects of soil properties and plant functional traits on soil bacterial and fungal diversity, and their relative contributions. Effect sizes are standardized coefficients for each predictor variable. Solid and open circles indicate significant and insignificant effects, respectively. *: $P < 0.05$; **: $P < 0.01$; ***: $P < 0.001$. RN is root nitrogen content, D is root depth, RTD is root tissue density, LC is leaf carbon content, LCN is leaf carbon nitrogen ratio, SLA is specific leaf area. The detailed description of each soil properties and plant functional traits is presented in Table S1.

Re: mSystems00570-25R1 (Soil properties and plant functional traits have different importance in shaping rhizosphere soil bacterial and fungal communities in a meadow steppe)

Dear Dr. Xian Yang:

Thank you for these last modifications.

Your manuscript has been accepted, and I am forwarding it to the ASM production staff for publication. Your paper will first be checked to make sure all elements meet the technical requirements. ASM staff will contact you if anything needs to be revised before copyediting and production can begin. Otherwise, you will be notified when your proofs are ready to be viewed.

Sincerely,
Juliana Almario
Editor
mSystems